# Application of Low-Cost Sensors for Accurate Ambient Temperature Monitoring

**Behnam Mobaraki [1,\*], Seyedmilad Komarizadehasl [2], Francisco Javier Castilla Pascual [3]**
**and José Antonio Lozano-Galant [1]**

1 Department of Civil and Building Engineering, Universidad de Castilla-La Mancha (UCLM),
 Av. Camilo Jose Cela s/n, 13071 Ciudad Real, Spain
2 Department of Civil and Environment Engineering, Universitat Politècnica de Catalunya Barcelona
 Tech (UPC), C/Jordi Girona 1-3, 08034 Barcelona, Spain
3 Department of Civil and Building Engineering, Escuela Politécnica de Cuenca, Campus Universitario,
 Universidad de Castilla-La Mancha (UCLM), 16071 Cuenca, Spain
\* Correspondence: behnam.mobaraki@uclm.es

**Abstract:** In structures with reduced monitoring budgets, the high cost of commercial metering devices is always an obstacle for monitoring structural health. This might be an issue when temperatures must be measured for both structural and environmental reasons. To fill this gap, in this paper, a novel monitoring system is proposed for the accurate measurement of indoor temperature in buildings. This protocol is characterized by its generality, as it can be easily adapted to measure any structural or environmental parameters on site. The proposed monitoring system uses from one to eight low-cost sensors to obtain multiple measurements of the ambient temperatures. The accuracy ranges of the developed monitoring systems with different numbers of sensors are statistically analysed. The results indicate that the discrepancy of the measurements decreases with the increase in the number of sensors, as the maximum standard deviation of 10 sensors (0.42) decreases to 0.32 and 0.27 for clusters of 20 and 30 sensors, respectively.

**Keywords:** building monitoring; Arduino microcontroller; thermal analysis; low-cost sensor; statistical analysis; indoor temperature

## 1. Introduction

The effects of extreme natural hazards (such as storms, earthquakes and climate change) on structures, together with natural deterioration, might require continuous evaluation and monitoring of them during their service life. Using different equipment, structural health monitoring (SHM) enables not only assessing serviceability of structures but additionally enables the detection and quantification of damage [1]. A number of SHM techniques have been proposed in the literature for monitoring various parameters of structures such as stresses, strains, accelerations and temperatures [2–6]. One of the main problems in the application of current systems is the high cost of the measuring devices, which can limit their applicability in structures with low monitoring budgets. As an alternative to traditional monitoring devices, a number of scholars have proposed low-cost solutions based on low-cost sensors [7] (LCSs) and open-source microcontrollers such as Arduino UNO [8], Arduino Nano [9], Arduino DUE [10], Arduino MEGA 2560 [11] and Raspberry [12]. Application of low-cost sensors for various monitoring projects can be found in the literature. For instance, Alvarez et al. [13] developed a real-time monitoring system for water content in cement mortars during the hydration process. Frei et al. [14] proposed a wireless sensor network to estimate building performance by measuring environmental parameters. Bidgoli et al. [15] used low-cost accelerometers to measure road pavement roughness. Rashid et al. [16] developed intuitive protocols to interact with electric appliances in smart buildings. Dave et al. [17] provided a web-based system that captures

information about energy usage, occupancy and user comfort through various types of sensors. An example of studying dynamic behaviour of scaled structures using low-cost accelerometer can be found in [18]. Lucchi et al. developed a low-cost and accurate conservation remote sensing technology for the hygrothermal evaluation of historic walls before and after retrofitting [19]. This methodology provided the assessment of the real condition of walls and to study various thermal insulation materials placed on the inner side. The proposed system was composed of sensor measurements, data acquisition system, data storage, and communication system. Aiming to lessen the cost, the data acquisition system was established based on Raspberry Pi 3. In addition, Amphenol temperature humidity sensors were used as they were characterized by high quality (RH: $\pm 2\%$ T: $\pm 0.3\,^{\circ}$C), a lower cost than the similar ones in the market, and easier acquisition compared with industrial ADC systems. Therefore, establishment of the proposed system required a smaller budget (EUR 20) compared to the commercial ones in the market (EUR 96). Given to the impossibility of removing samples and the unknown hygrothermal behavior of walls, the same authors developed an in situ hot box providing analysis of various insulation technologies when applied on a historic building so as to determine the hygrothermal performance of historic masonry buildings [20]. A systematic literature review of application of LCSs for building monitoring has been presented in the article of Mobaraki et al. [21].

Figure 1 presents the evolution of articles published corresponding to the use of low-cost sensors in various fields, from 2011 to 2021. The information visualized in this figure is in percentages, which correspond to the quantity of publications in the associated year relative to the whole number of publications in the studied period. Figure 1 illustrates this information corresponding to the fields of civil and building engineering, medical engineering, electrical engineering, industrial engineering, mechanical engineering, and architecture.

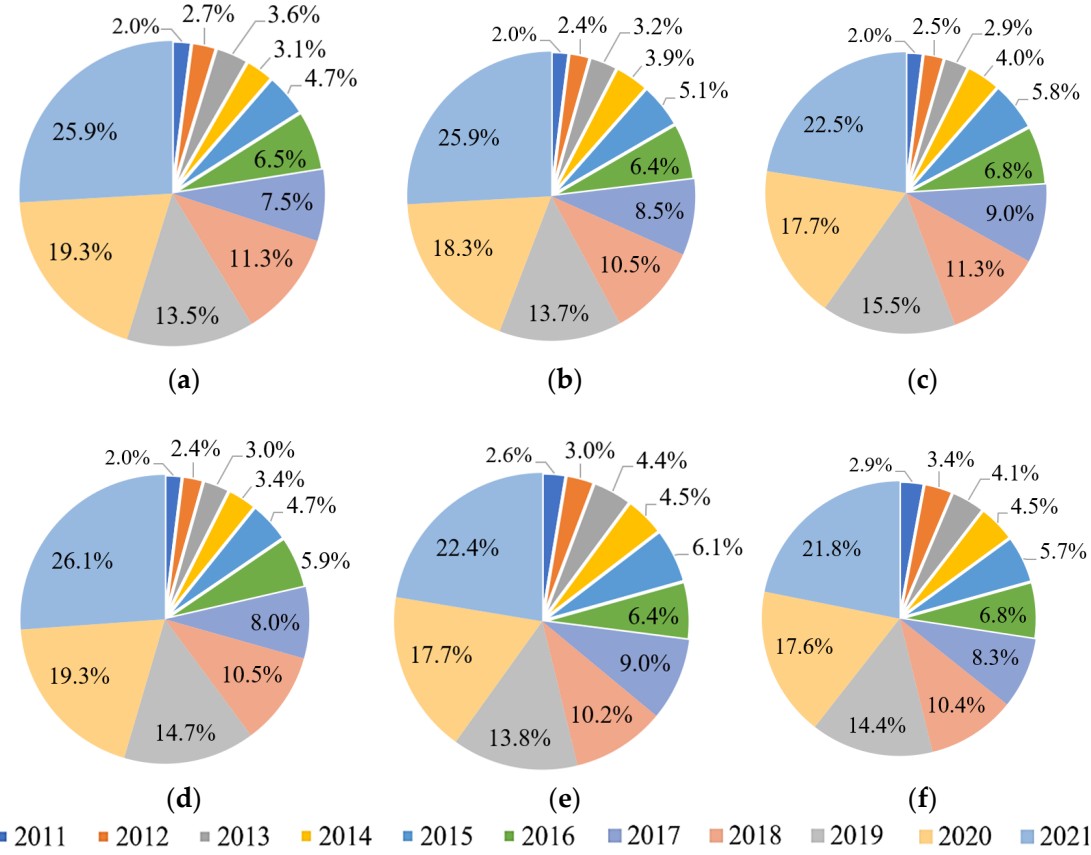

**Figure 1.** Evolution of studies referenced in Scopus database dealing with low-cost sensors, from 2011 to 2021: (**a**) civil–building; (**b**) medical; (**c**) electrical; (**d**) industrial; (**e**) mechanical; (**f**) architecture.

The strategy used to obtain the information in Figure 1 is based on the following two steps. (1) Using keywords for various fields: a number of keywords such as "low-cost monitoring", "low-cost sensor", "monitoring", and "SHM" were introduced in the Scopus database. The obtained results were filtered in the following fields: civil and building engineering, industrial engineering, mechanical engineering, medical engineering, electronic engineering, and architecture. This step was followed by the search algorithm "AND". (2) Filtering the obtained articles: The obtained articles were filtered to eliminate duplications and out of topic works. This filtration led to a massive reduction in the quantity of articles found in the first step. For instance, considering the field of mechanical engineering only in 2021, the number of articles was reduced from 4753 to 1867. In the same way, the entire number of publications found in the field of civil and building engineering was reduced from 3052 to 2334. In this step, the search algorithm "AND NOT" was adopted for filtration. The remaining quantity of the articles was clearly addressing or assessing the application of low-cost sensors in the studied disciplines. It is also relevant to state that in this search algorithm, the authors were not focusing only on the English language articles. Thus, the acquired publications contain various languages from different parts of the world. The reviewed methodology was followed by another work in the literature [22] and by the same authors in the same journal (Buildings) [21]. Environmental temperature plays an important role in controlling a number of phenomena on sites as well as in the analysis of energy consumption in buildings [23]. Monitoring is commonly limited by equipment availability; therefore, elements and spaces are traditionally represented by a single temperature value. Connection problems and unexpected circumstances may invalidate long-term monitoring records when using only one device. The advantages of using several low-cost sensors have been stated in a number of studies [24]. For example, the authors developed a novel data acquisition system using low-cost sensors, to characterize thermal parameters of building envelopes [25]. Additional literature examples of the application of multiple low-cost sensors for environmental monitoring purposes are summarized in Table 1. This table includes the proposed application (air quality, ambient monitoring and indoor environmental monitoring), the sensor types, and the number of sensors used in each of the reported works.

**Table 1.** Previous studies using multiple low-cost sensors for environmental monitoring.

| Application | Sensor Type | Number of Sensors | Reference |
|---|---|---|---|
| Air quality | $CO$, $NO$, $NO_2$ | 4 | [26] |
| Air quality | $MO_x$, SHT21, light sensor | 2 | [27] |
| Air quality | $CO$, $NO$, $O_3$, $NO_2$ | 5 | [28] |
| Indoor environmental monitoring | SHT15-NTC-TSLl2561-PARALLAX-SENSAIR K30- | 2 | [29] |
| Indoor environmental monitoring | DHT11-DS18B20-LM35 | 3 | [30] |
| Ambient $NO_2$ monitoring | Alphasense cell | 16 | [31] |

All of the studies in Table 1 define the monitoring accuracy with multiple sensors. Nevertheless, none of these works studied the statistical benefits of increasing the number of sensors in the monitoring of ambient temperatures. To fill this gap, this paper presents a novel monitoring system using cheap sensors available in the market. Unlike commercial thermometers, this protocol is based on multiple measurements of the indoor temperatures, enabling the application of a statistical analysis to improve the low accuracy of the low-cost sensors. This monitoring system was applied to five of the low-cost sensors most commonly used in the literature (BMP280, BMP180, DHT22, SHT21 and SHT35) to develop five independent measurement devices. For each of these devices, the improvements in accuracy by increasing the number of sensors (ranging from 1 to 8) were studied. The five developed devices were calibrated with statistical references obtained by averaging the measurements of 30 sensors of each studied sensor type. This monitoring system is characterized by its generality; it can be applied to measure any structural (such as stress,

strain or acceleration) or environmental (such as transmittance value, resistance parameter, rate of heat flux or conductivity coefficient) parameters on site. Monitoring of these parameters is a decisive factor for maintenance operations of buildings [32], bridges [33], tunnels [34,35], and other infrastructure [36]. Implementation of the proposed methodology helps engineers to improve maintenance activities and operational efficiency while at the same time decreasing the cost and increasing the accuracy [37]. Application of affordable and reliable monitoring systems could solve challenges of long-term monitoring of the aforementioned structures [38].

This paper is organized as follows. In Section 2, the hardware and software of the five developed monitoring system are detailed. In Section 3, the main characteristics of the commercial thermometers in the literature are reviewed. In Section 4, a laboratory test is described that compares the accuracy of the developed monitoring system with thermometer uses in the literature. Then, the normal distribution of the data of the sensor sets is demonstrated. Next, the benefits of increasing the number of low-cost sensors are studied to provide recommendations for the selection of the types and numbers of sensors for different accuracy ranges. A cost comparison between the proposed monitoring systems and commercial alternatives is also presented. Finally, in Section 5, some conclusions are drawn.

## 2. Development of a Novel Monitoring System

In this section, the hardware and software of the five proposed monitoring system devices are presented.

### 2.1. Hardware of the Proposed Monitoring System

Various types of low-cost sensors are available to derive the temperature of surroundings. Examples of these sensors are the DHT11 [39], DHT22 [40], SHT10 [41], SHT21 [42], SHT35 [43], BMP180 [44], BMP280 [45], and LM75 [46]. The selection among these alternatives is traditionally based on the information presented in the commercial datasheets. The main characteristics of the abovementioned sensors (model, detection range, accuracy, resolution, response time, communication protocol, and cost) are listed in Table 2, which includes references to the application of associated sensors in the literature.

**Table 2.** Specifications of low-cost thermometers from their catalogues.

| N° | Model | Detection Range (°C) | Accuracy (°C) | Resolution (°C) | Response Time (s) | Communication Protocol | Cost (EUR) | Reference |
|---|---|---|---|---|---|---|---|---|
| 1 | DHT11 | [0 to 50] | 2 | 0.1 | 2 | Single wire/bus | 1.56 | [39] |
| 2 | DHT22 | [−40 to 80] | 0.5 | 0.1 | 2 | Single wire/bus | 5.40 | [47] |
| 3 | SHT10 | [−40 to 125] | 0.5 | 0.01 | 8 | $I^2C$ | 4.57 | [41] |
| 4 | SHT21 | [−40 to 125] | 0.3 | 0.01 | 8 | $I^2C$ | 4.61 | [42] |
| 5 | SHT35 | [−40 to 125] | 0.2 | 0.01 | 8 | $I^2C$ | 5.76 | [43] |
| 6 | BMP180 | [−40 to 85] | 2 | 0.1 | 0.0075 | $I^2C$ | 3.72 | [44] |
| 7 | BMP280 | [−40 to 85] | 1 | 0.01 | 0.55 | $I^2C$ and SPI | 3.59 | [45] |
| 8 | LM75 | [−55 to 125] | 1 | 0.1 | 0.5 | $I^2C$ | 2.80 | [46] |

This paper studied the five most commonly used sensors, as shown in Table 2 (BMP180, BMP280, DHT22, SHT21 and SHT35). The chosen sensors for establishment of the five monitoring systems have different characteristics in terms of the accuracy, response time, communication protocol, and cost. The accuracies of these sensors range between 0.2 °C (SHT35) and 2 °C (BMP180), while their prices range between EUR 3.59 (BMP280) and EUR 5.76 (SHT35). The sensors studied are based on different communication protocols. For example, the sensor DHT22 uses a single wire/bus, while BMP180, SHT21 and SHT35 operate with the inter-integrated circuit protocol, $I^2C$, and sensor BMP280 uses both $I^2C$ and the serial peripheral interface (SPI) [48].

An independent monitoring system was built with each of the five selected low-cost sensors. Each of these monitoring systems included the following elements. (1) The Arduino MEGA 2560: This microcontroller enables the physical computing and controlling of the protocol. This device was previously used in other complex projects (such as [30,31]). (2) A power source: A USB cable was attached to the laptop to supply the required power for an individual monitoring system. (3) The multiplexer TCA9548A: for enhancing the amount of network data transferred in a specific period of time by sensors with $I^2C$ communication protocols. (4) Breadboard: for easing the assembling of the system wires. (5) A clock sensor DS3231: for capturing the exact date and time of each measurement. (6) An SD card: for storing the data of each individual. Finally, (7) low-cost sensors.

Each monitoring system included 30 of the following sensors: BMP180, BMP280, DHT22, SHT21 and SHT35. To provide a better understanding of the proposed methodology (statistical reference as well as combinatorial analysis), Figure 2a,b are presented. Figure 2a depicts the defined order/label of the installed sensors for each of the five monitoring systems in the written algorithm. The statistical reference of each monitoring system comes from the mean value of the associated 30 sensors. Figure 2b demonstrates the order of the sensor's selection for combinatorial analysis, ranging from 1 to 8 out of 30 sensors. Through this combinatorial analysis, all the possible sensor arrangements will be chosen and compared with the associated statistical reference.

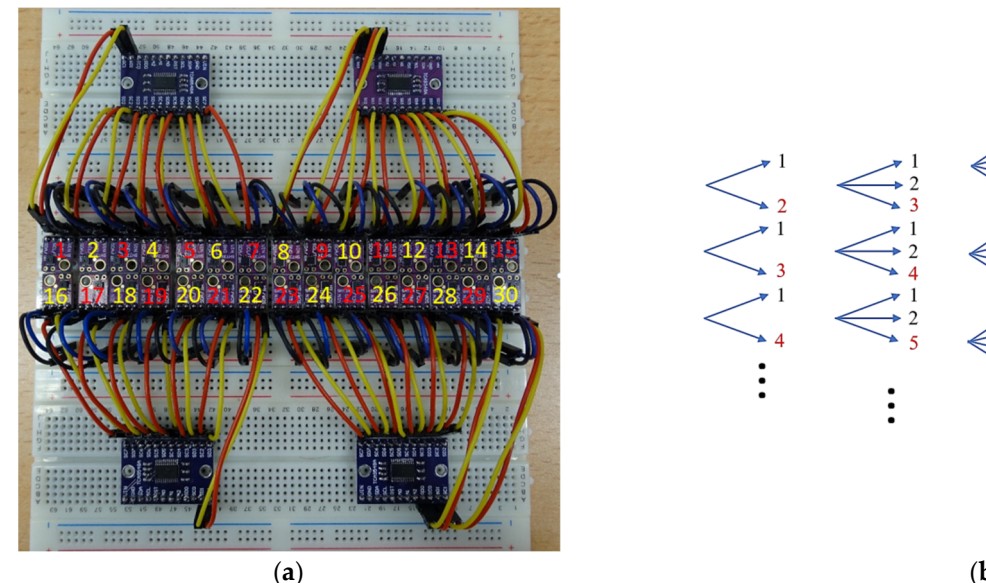

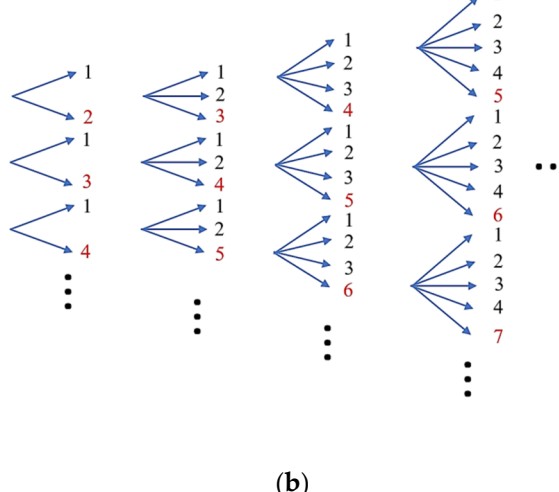

(**a**)          (**b**)

**Figure 2.** The defined sensor configuration for each monitoring system in the algorithm (**a**) and the order of the sensor selection for combinatorial analysis (**b**). The numbers in this Figure refer to the order of the sensors for selection of various sensor arrangements in combinatorial analysis.

Table 3 provides information over the main characteristics (accuracy, response time and detection range) of the five developed data acquisition systems. This table also includes a picture of an individual developed monitoring system and the sketch of the wiring connections of a single sensor to Arduino MEGA using Fritzing software [49]. The analysis of this table shows that the utilized sensors were characterized by different accuracies, response time, and detection ranges. This use of different models of sensors provides the opportunity to investigate the statistical benefits of increasing the number of sensors on different sensor types. As shown in Table 3, four breadboards were used to ease the connections of the 30 sensors of an individual system. The size of each monitoring system was 27 × 6 cm. These dimensions are close to those of other commercial thermometers in the literature (such as TESTO 435-1 and FLUKE 971, with 22 × 7.5 cm [50] and 19.5 × 6 cm [51], respectively).

**Table 3.** Established data acquisition systems and the associated technical characteristics of the used sensors.

| System | Accuracy (°C) | Response Time (s) | Detection Range (°C) | Figure | Wire Connection to Arduino MEGA |
|--------|---------------|-------------------|----------------------|--------|----------------------------------|
| BMP280 | 1.0 | 0.5500 | −40 to 85 |  |  |
| BMP180 | 2.0 | 0.0075 | −40 to 85 |  |  |
| DHT22 | 0.5 | 2.0000 | −40 to 80 |  |  |
| SHT21 | 0.3 | 8.0000 | −40 to 125 |  |  |
| SHT35 | 0.2 | 8.0000 | −40 to 125 |  |  |

### 2.2. Software

For the development of each monitoring system, two different algorithms were written for the following purposes: (1) capturing the data from low-cost sensors and (2) analysing the obtained data from an individual monitoring system. These algorithms are detailed in the following sections.

### 2.2.1. Algorithm for Reading Sensor Data

Arduino IDE was used for programming the functionality of the input data introduced to the ports of the microcontrollers in each of the five developed monitoring systems. This algorithm was divided into the following steps. (1) Loading the SD card library (SD.h), the serial port interface library (SPI.h) and the corresponding sensor library (BMxI2C.h) into the project sketch. (2) Initializing the serial monitor and ports of the multiplexers to enable the readings. (3) Adjusting the measuring frequency and opening the SD card for saving the temperature measurements. In all the measurement sets, the recording intervals were set as the minimum response time of the commercial device, EL-USB-2 LASCAR (10 s). (4) Recording the temperatures and saving them on the SD card over time.

A summary of the developed algorithm is presented in Figure 3. To ease the algorithm replication, specific Arduino functions are included on the right of this figure.

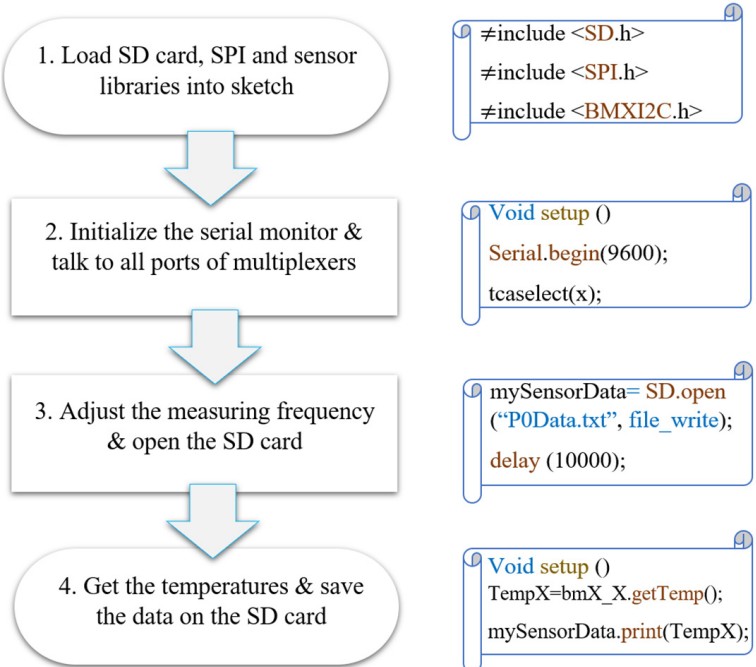

**Figure 3.** Algorithm to record the temperature measurements on each monitoring system.

### 2.2.2. Data Analysis Algorithm

To establish a statistical reference, each monitoring system included 30 sensors whose temperature outputs were monitored with the algorithm presented in the previous section. This statistical reference is independent for each monitoring system and is obtained by averaging the measurements of all the sensors. The different statistical references were used to evaluate the accuracy improvements in each monitoring system when increasing the number of sensors. To define the number of possible combinations for a specific number of sensors, the following combinatory equation was followed:

$$C(n,r) = (n!)/(r!(n-r)!) \tag{1}$$

where *n* is the number of sensors (30 sensors) and *r* the number of sensors used in each combination (after checking the results, this number ranged from 1 to 8). To obtain the arrangements of the different numbers of sensors, an algorithm was developed in MATLAB. "Arrangement" is defined as each of the sensor configurations presented in Figure 2b. The main steps of this algorithm for the calibration and data analysis are as follows: (1) input the data from the SD card, (2) establish the statistical "reference temperature" from the average of the measurements of the 30 sensors of each individual monitoring systems, (3) input the desired number of sensors to be considered in each combination, determine the possible sensor arrangements and the associated mean values, and finally compare them with

the statistical reference temperature so as to check the performance of the chosen sensor arrangement, (4) select both the most accurate and inaccurate sets of sensors, and (5) check the repeatability of the sensors through the standard deviation. A summary of these steps is shown in Figure 4. With the purpose of specifying the most accurate/inaccurate sets of arrangements, the order of the sensors in all monitoring systems has been specified in the proposed algorithm. To evaluate the accuracy of the sensor arrangements, two parameters were compared with the reference temperatures of each monitoring system over time: (1) average temperature and (2) standard deviation. This algorithm enables the automatic identification of outliers.

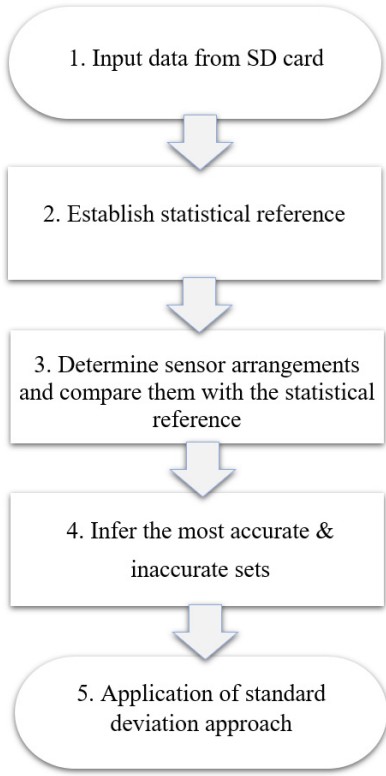

**Figure 4.** Algorithm in MATLAB to analyse the recorded data.

The number of possible arrangements for the different sensors studied (from 1 to 8) obtained by this algorithm is summarized in Table 4, together with the computation times. Computation time refers to the duration required for conducting the whole loop presented in Figure 4. Statistical analysis of the sensor arrangements provides the accuracy range (minimum and maximum errors) of the different sensor configurations in each monitoring system.

**Table 4.** Number of arrangements and computation time for all possible configurations of sensors in monitoring system with 30 sensors.

| N° of Sensors | N° of Arrangements | Time (min) |
|:---:|:---:|:---:|
| 1 | 30 | 0.01 |
| 2 | 435 | 0.01 |
| 3 | 4060 | 0.02 |
| 4 | 27,405 | 0.09 |
| 5 | 142,506 | 0.50 |
| 6 | 593,775 | 4.28 |
| 7 | 2,035,800 | 19.22 |
| 8 | 5,852,925 | 28.16 |

## 3. Commercial Thermometers

In this section, six of the commercial thermometers most commonly used in the literature to monitor ambient temperatures are studied. The main characteristics of these devices are listed in Table 5. This table includes the model's name, the sensor accuracy, the temperature range, price, and the associated references to the application of the instrument in the literature.

**Table 5.** Commercial thermometers in the market and literature.

| N° | Model | Accuracy (°C) | Range (°C) | Price (EUR) | Reference |
|---|---|---|---|---|---|
| 1 | PROTMEX MS6508 | 1.0 | [−20 to 60] | 50 to 60 | [52] |
| 2 | REED R6001 | 0.8 | [−20 to 60] | 130 to 150 | [53] |
| 3 | FLUKE 971 | 0.5 | [−20 to 60] | 350 to 500 | [54] |
| 4 | EL-USB-2 LASCAR (*) | 0.5 | [−35 to 80] | 50 to 100 | [55] |
| 5 | TESTO 435-3 (**) | 0.2 | [−25 to 75] | 750 to 1200 | [56] |
| 6 | EXTECH EN510 | 0.1 | [−100 to 1300] | 180 to 220 | [57] |

(*) Price can vary depending on the number of items ordered. (**) Including external probes for air and surface temp.

An analysis of Table 5 shows the commercial sensor accuracy ranges between 0.1 °C (EXTECH EN510) and 1.0 °C (PROTMEX MS6508). The price range presents significant variations between the cheapest sensor (PROTMEX MS6508 with a cost of EUR 57) and most expensive one (TESTO 435-1 with a cost of EUR 750).

A commercial thermometer with intermediate characteristics (EL-USB-2 LASCAR thermometer) was used simultaneously to validate the performance of the five developed monitoring systems. This device works with a normal 1/2 AA battery and can export its data to a PC through a USB port.

## 4. Experiment

This section describes an experimental test conducted to validate the performance of the five developed monitoring systems. First, the experiment is described, and the obtained results are compared. Then, the normal distribution of the data recorded by the five studied sensor types is evaluated. Finally, the accuracy range (maximum and minimum errors) with different numbers of sensors in each monitoring system is statistically studied.

### 4.1. Description

The ambient temperatures measured by the five developed monitoring systems were compared with those obtained simultaneously by the commercial thermometer EL-USB-2-LASCAR. Different devices used in the experiment are illustrated in Figure 5. The experiment was conducted in a controlled environment in the laboratory of the Civil Engineering Department of the University of Castilla-La Mancha (UCLM) in Ciudad Real, Spain in February 2020. During the experiment, the temperature was controlled by the air conditioning (AC) and divided into the following two scenarios: Scenario 1, in which the temperature was kept constant at 25 °C, and Scenario 2, in which the AC was turned off with the consequent drop in the temperature. The sensor locations were chosen far from windows to avoid the effects of direct sun radiation.

During the experiment, the laboratory temperature was recorded for 12 h continuously, with a sampling frequency of 10 s, providing more than 4000 readings in each of the 30 sensor copies of the five monitoring systems as well as in the commercial thermometer.

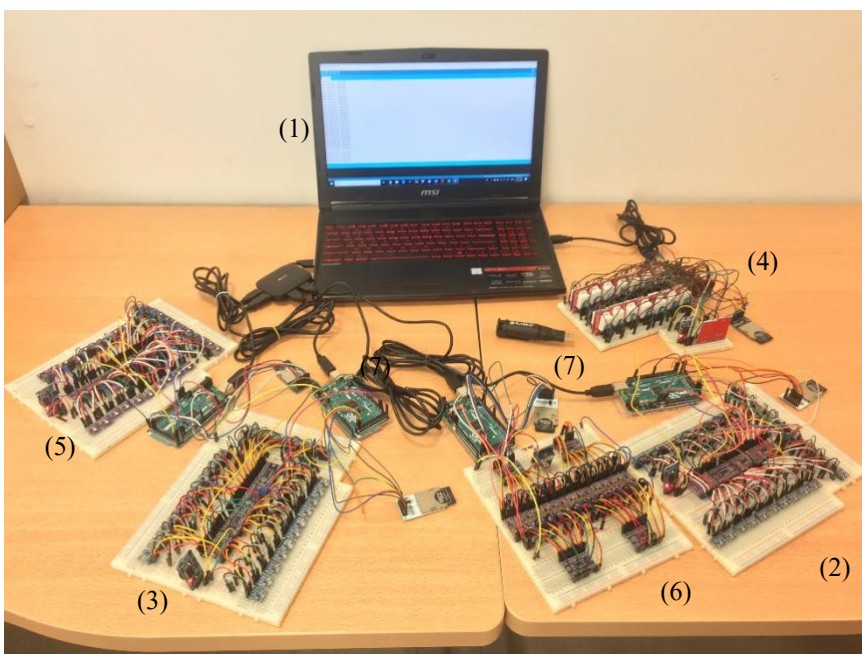

**Figure 5.** Elements in the experiment: (1) laptop, (2) monitoring system BMP280, (3) monitoring system BMP180, (4) monitoring system DHT22, (5) monitoring system SHT21, (6) monitoring system SHT35, and (7) EL-USB-2-LASCAR.

### 4.2. Analysis of the Reference Temperatures

After applying the algorithm of Section 2.2.1 for sensor data reading, the ambient temperatures of each sensor was recorded. Figure 6 plots the evolution of the temperature recorded by each of the 30 copies of sensor SHT35 obtained in the corresponding data acquisition system. This figure clearly demonstrates the synchronised measurements of the SHT35 sensors to identify the laboratory temperatures in the two studied scenarios.

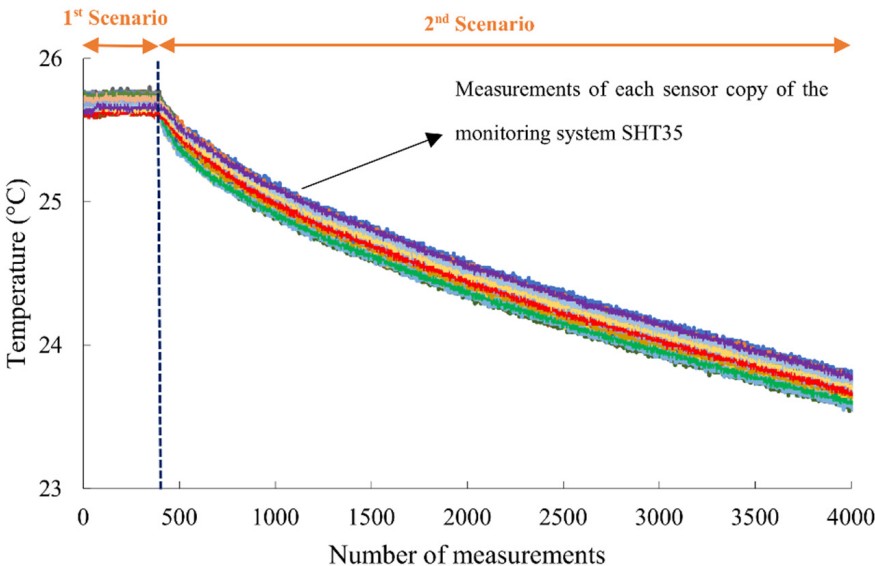

**Figure 6.** Evolution of the temperatures recorded by the 30 sensor copies of SHT35.

Dispersion of the measurements of the five developed monitoring systems is presented in Figure 7, in terms of the standard deviations of the reference temperatures (average of the 30 sensor copies) over time.

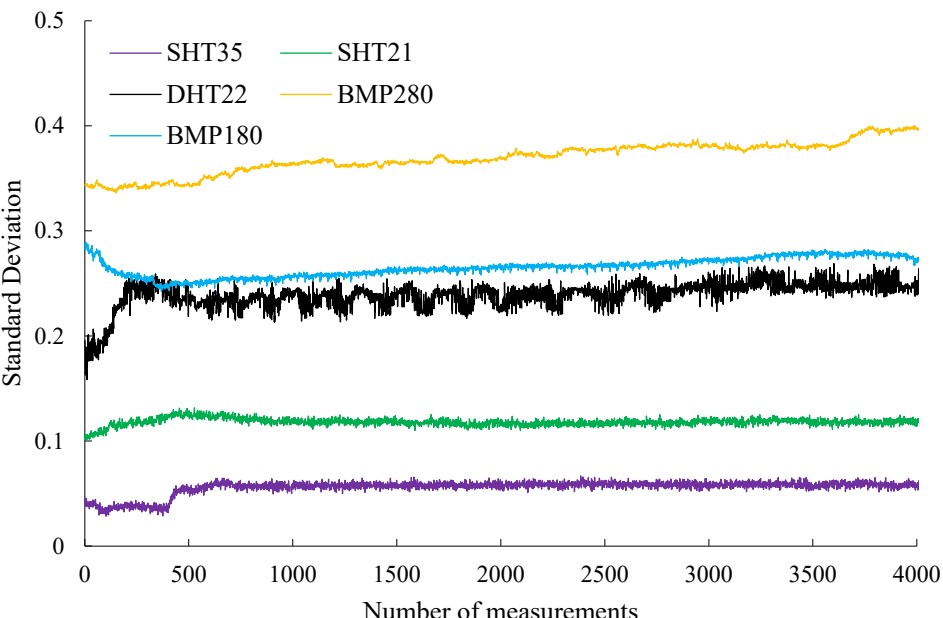

**Figure 7.** Standard deviations of the five monitoring systems with 30 sensors.

The analysis of Figure 7 illustrates that different standard deviation ranges are obtained for the different types of sensors analysed. From these results, it can be concluded that sensors SHT35 and BMP280 have the lowest ($0.028 \leq SD_{SHT35} \leq 0.066$) and the highest ($0.336 \leq SD_{BMP280} \leq 0.404$) standard deviation ranges, respectively. Precise assessment of this figure additionally indicates that temperature sensors (such as SHT21, SHT35 and DHT22) have better performance in dynamic environments (Scenario 2 in the last 3600 measurements) than in the static ones (Scenario 1 in the first 400 measurements). During Scenario 1, the sensor nodes recorded scattered values, and the associated standard deviations gradually increased. Furthermore, after encountering changes of temperature (Scenario 2), the standard deviation remained practically constant with limited fluctuations. In the case of the pressure sensors (such as BMP180 and BMP 280), the standard deviation changed gradually in both the static and dynamic scenarios and did not follow any specific pattern.

Standard deviation (SD) is a measure to determine the dispersion of data relative to its average value where a higher range of this parameter presents a higher uncertainty of measurements. Figure 8 studies the impact of increasing number sensors on variation of SD, randomly for one of the monitoring systems, DHT22. To do so, this figure plots the ranges of the SD when the number of sensors was increased between 10 and 30 units. In this figure, the vertical axis refers to the obtained SD, while the horizontal one refers to the measurements taken in the test.

According to Figure 8, the uncertainty of measurements was lessened by enhancing the number of sensors. Increasing the number of sensors resulted in a decreasing trend of standard deviation of the monitoring system DHT22 as: $0.19 \leq SD_{10\ sensors} \leq 0.42$, $0.18 \leq SD_{20\ sensors} \leq 0.32$, and $0.16 \leq SD_{30\ sensors} \leq 0.27$. In other words, the standard deviation related to the observations of the most inaccurate 20 sensors tend to be closer to the associated statistical reference than that of the 10 most inaccurate sensors.

To assess the applicability of the five developed monitoring systems, a comparison between the reference temperatures and the commercial thermometer EL-USB-2 LASCAR is presented in Figure 9. This Figure illustrates the time history of the temperature captured by the EL-USB-2 LASCAR and the reference temperature of each monitoring system.

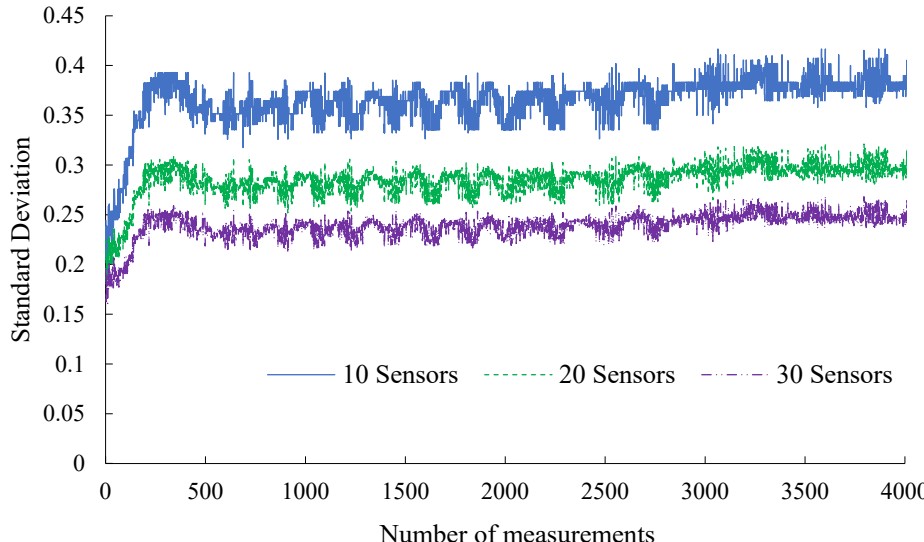

**Figure 8.** Standard deviation of the monitoring system DHT22 with different numbers of sensors (10, 20 and 30).

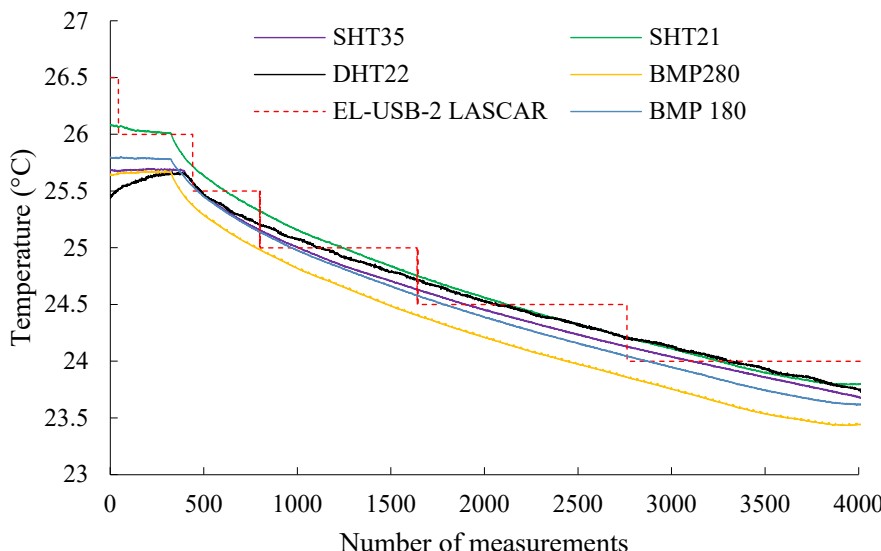

**Figure 9.** Comparison between the reference temperatures recorded by the five monitoring systems with measurements of the commercial thermometer EL-USB-2 LASCAR.

The analysis of Figure 9 shows significant differences between the results of EL-USB-2 LASCAR and the reference temperatures of the developed monitoring systems. The limited accuracy and resolution of the EL-USB-2 LASCAR device (0.5 °C) resulted in a sawtooth-shaped temperature plot. In contrast, the developed monitoring systems recorded continuous increments of temperature that were closer to the actual temperature variation.

To further study differences between the measurements of the developed monitoring systems and the EL-USB-2 LASCAR, Figure 10 is presented where the error of this commercial thermometer is calculated in terms of the statistical reference of an individual system. An overall result of this figure is that the lower the range of temperatures captured by the monitoring systems (according to the Figure 9), the higher the rate of errors of EL-USB-2 LASCAR. According to the obtained results, the highest and the lowest ranges of errors were obtained when considering the statistical references of the monitoring systems BMP280 and SHT21, respectively.

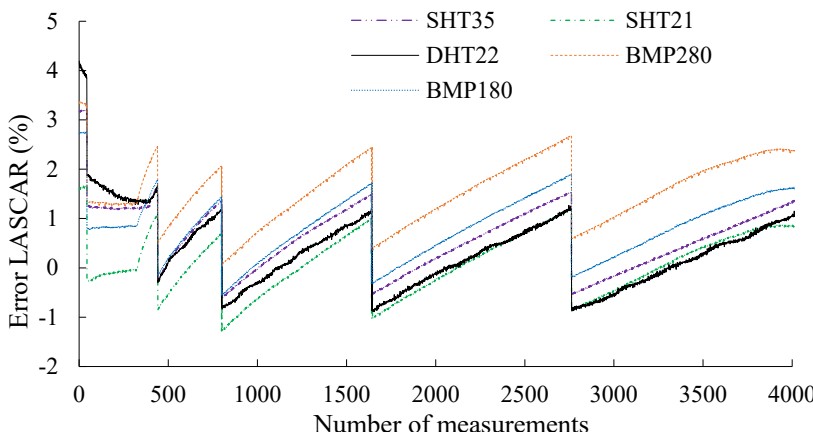

**Figure 10.** Errors of EL-USB-2 LASCAR calculated from the reference temperatures of the five developed monitoring systems.

### 4.3. Normality Test of the Reference Temperature

A statistical normal-distribution analysis (normality test) of the data obtained by the five developed monitoring systems was conducted to verify whether the temperatures measured by the different sensor copies follow a normal distribution. This normality test was performed for random clusters of the monitoring information using the software SPSS-2019 [58]. Table 6 summarizes the statistical values of one of the analysed clusters in each of the five developed monitoring systems. The results in this table include the mean, standard deviation (SD), minimum (Min) and maximum (Max) recorded temperatures, standard error of skewness (Std. Error. Sk), standard error of kurtosis (Std.Err.Ku), Z-value skewness (Z-val.Sk), and Z-value kurtosis (Z-val.Ku).

**Table 6.** Statistical data analysis in SPSS for each developed monitoring system.

| | **Monitoring System** | | | | |
|---|---|---|---|---|---|
| | **SHT35** | **SHT21** | **DHT22** | **BMP280** | **BMP180** |
| Mean | 25.68 | 26.08 | 25.43 | 25.64 | 25.79 |
| SD. | 0.04 | 0.11 | 0.19 | 0.35 | 0.29 |
| Min. | 25.60 | 25.84 | 25.00 | 25.18 | 25.18 |
| Max. | 25.76 | 26.28 | 25.90 | 26.35 | 26.28 |
| Std.Err.Sk | 0.43 | 0.43 | 0.43 | 0.43 | 0.427 |
| Std.Err.Ku | 0.83 | 0.83 | 0.83 | 0.83 | 0.83 |
| Z-val.Sk | −0.65 | −0.57 | −0.37 | 1.07 | −0.74 |
| Z-val.Ku | −1.08 | −0.42 | −0.05 | −1.40 | −0.81 |

As indicated by [59], skewness (Z-val.Sk) and kurtosis (Z-val.Ku) Z-values should be in the range of −1.96 to 1.96. As shown in Table 6, according to the inferred Z-values, it can be expressed that all the five clusters of data are normally distributed in terms of skewness and kurtosis since they are within a range of +/−1.96.

The normal distribution of the measurements of the 30 copies of the five developed monitoring systems is illustrated in Figure 11. The Q-Q plots in Figure 11a,d,g,j,m show that all monitoring systems follow a normal distribution, as their data fall well along the expected regression line. The sensor that provided the worst approximation was BMP280. The histograms in Figure 11b,e,h,k,n illustrate the Gaussian distribution via the bell-shaped curve of the different recorded data associated with the monitoring systems SHT35, SHT21, DHT22, BMP180, and BMP80, respectively. These figures visually assess skewness, anomalies and frequency of observations. Due to the minimum discrepancy of the measurements obtained from SHT35 and SHT21, the distributions in histograms are symmetric. However, sensor BMP280 presents a higher asymmetry. Finally, the box-and-whiskers plots

in Figure 11c,f,i,l,o provide the overall information over the obtained temperature ranges captured from SHT35, SHT21, DHT22, BMP180, and BMP80, respectively. The sensor with the narrowest range was the SHT35 ($25.76 - 25.6 = 0.16$ °C), while the sensor with the widest range was the BMP280 BMP280 ($26.35 - 25.18 = 1.2$ °C). The box-and-whiskers plot related to the DHT22′ set indicates that this metering system has the maximum number of observations close to the median. However, it contains the minimum number of temperatures detected on the top 25% phase of the records.

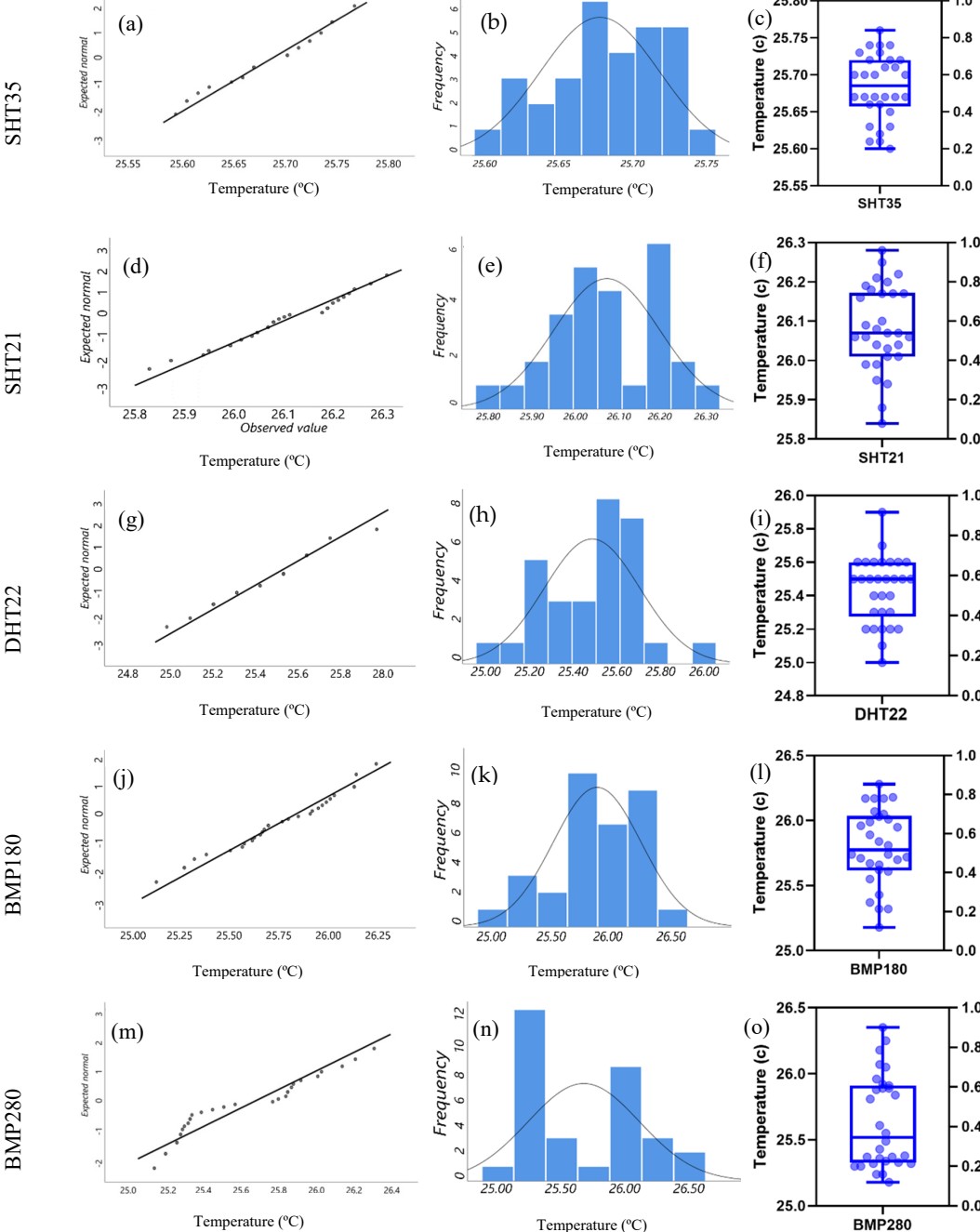

**Figure 11.** (**a**) Normal Q-Q plots (SHT35), (**b**) Histograms (SHT35), (**c**) Box-and-whiskers plot (SHT35), (**d**) Normal Q−Q plots (SHT21), (**e**) Histograms (SHT21), (**f**) Box-and-whiskers plot (SHT21), (**g**) Normal Q−Q plots (DHT22), (**h**) Histograms (DHT22), (**i**) Box-and-whiskers plot (DHT22), (**j**) Normal Q−Q plots (BMP180), (**k**) Histograms (BMP180), (**l**) Box-and-whiskers plot (BMP180), (**m**) Normal Q−Q plots (BMP280), (**n**) Histograms (BMP280), and (**o**) Box-and-whiskers plot (BMP180).

### 4.4. Analysis of the Individual Sensors

The accuracy ranges of the five analysed sensor types are illustrated in Figure 12, where the errors of the most and least accurate sensors were derived and compared with the information presented in commercial catalogues of the associated sensor (presented in Table 2).

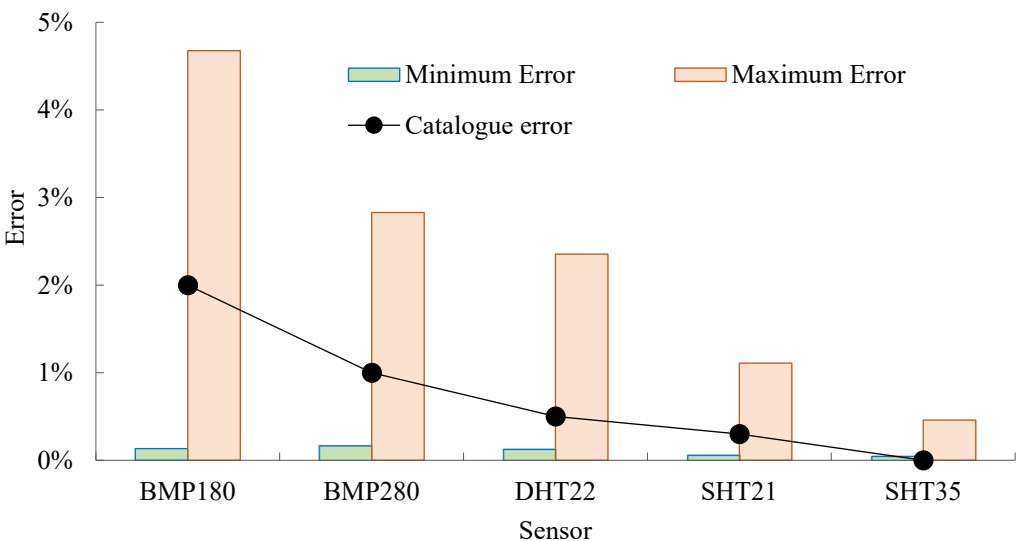

**Figure 12.** Comparison between the most- and least-accurate sensors in the sets and the accuracies described in their respective commercial catalogues.

Figure 12 shows that the higher the accuracy of the sensor is, the smaller both the maximum and minimum calculated errors are by the reference temperatures. This figure additionally illustrates that the information in the catalogues of the different sensor types does not correspond with the actual performance of these devices on site. For example, the maximum obtained error of sensor BMP180 (4.68%) was 57% higher than that presented in the catalogue for this sensor (2%).

### 4.5. Analysis of the Developed Monitoring Systems with Different Number of Sensors

The error of each single sensor is systematic and comes from a number of uncertainties (such as observational errors, instrumental errors and environmental errors). To reduce the measurement errors of the sensors in the different monitoring systems, their respective measurements were averaged with the data analysis algorithm presented in Section 2.2.2. Temperature analysis was conducted for each monitoring system. In this analysis, the average temperatures of all the possible arrangements of sensors, from one to eight, of each monitoring system were compared with the corresponding reference temperatures to obtain the accuracy ranges (maximum and minimum errors).

The maximum errors (related to the most inaccurate sensor arrangements) for the five studied monitoring systems were compared with those of state-of-the-art thermometers (from 0.1 to 0.5 °C), which are presented in Figure 13. In this figure, the horizontal axis represents the number of sensors in different sensor combinations (ranging from one to eight), while the vertical axis shows the absolute value of the maximum error (Abs(Max Error)) obtained by all possible combinations of the sensor arrangements described in Table 4. The errors in Figure 13 are calculated from the average measurements of the different sensor arrangements and the associated reference temperature, and they represent the maximum expected errors of a random set of sensors that are chosen. The area between the red arrows in the figure highlights the range of errors of the commercial devices presented in Table 5 (from 0.1 to 0.5 °C).

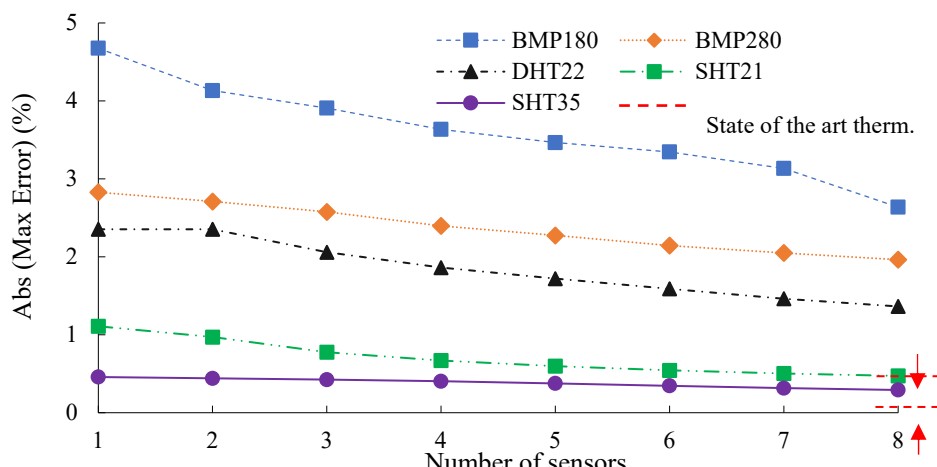

**Figure 13.** Maximum absolute errors of all possible arrangements of the different sensors in the five developed monitoring systems and in the state-of-the-art thermometers.

Figure 13 illustrates that the benefits of adding additional sensors depend to a great extent on the sensor type and its accuracy. In regard to sensors with lower accuracy (such as the BMP180), adding sensors presents higher benefits. For example, the maximum error obtained from measuring temperatures using a unique BMP180 sensor (4.67%) is reduced to 4.13% when two sensors are considered (11.6% reduction of the error). This error is reduced to 2.26% when eight sensors are considered. Furthermore, the sensors with higher accuracy (such as the SHT35) present lower marginal benefits. For example, in this case, the errors with one unique sensor copy (0.46%) are reduced to 0.44% when two sensor copies are considered (4.3% reduction of the error) and to 0.29% when eight sensor copies are considered.

The analysis of Figure 13 additionally illustrates how the maximal errors are reduced with more sensors. The higher the number of sensors averaged, the lower the maximum errors in the reference temperature. In the worst sensor combinations, the accuracy of the FLUKE 971 and EL-USB-2 LASCAR thermometers (0.5 °C) can be achieved when averaging eight SHT21 sensors (concretely, 0.47 °C). The same precision can be obtained directly with any of the studied numbers of the SHT35 sensor. Although the error reduction is not significant, the use of at least three of these sensors is advised to have multiple simultaneous temperature readings. In this case, an accuracy of 0.42 °C is obtained. In the case of the rest of the studied sensors (BMP180, BMP280 and DHT22), an average of more than eight copies is required to achieve an accuracy of 0.5 °C, considering the worst sensor combination.

Not all sensor types were easy to work with. During the assembling process, a few sensor types were easily burned out due to variations in the voltage in the circuit. To take this problem into account and to determine the real cost of the monitoring systems, a correction factor ($F_C$) was devised, determined by dividing the total number of sensors needed by 30 (number of sensors in each monitoring system). For example, in the case of sensor type SHT21, eight sensors were burned out during the assembling process such that 38 sensors were needed to achieve the targeted 30. Therefore, the correction factor of this type of sensor is $F_{C.SHT21} = 38/30 = 1.3$ . Assuming a linear response, this parameter can be used to include the effects of the burned-out sensors in the monitoring systems with any number of sensors. The $F_C$ values for the monitoring systems BMP180, BMP280, DHT22, and SHT35 are 1.1, 1.2, 1.0 and 1.0, respectively.

The costs of the two selected monitoring system alternatives (SHT21 with eight sensors and SHT35 with three sensors) with maximum errors are reviewed in Table 7. This table includes the prices of the monitoring system elements as follows: (1) sensors: this number is obtained by multiplying the sensors by the corresponding $F_c$ factor; (2) microcontroller; (3) breadboard; and (4) the multiplexer.

**Table 7.** Cost of the chosen monitoring systems with the least accurate sensors of SHT21 and SHT35.

| | SHT35 | | SHT21 | |
|---|---|---|---|---|
| **Components** | **Price (EUR)** | **Nº** | **Price (EUR)** | **Nº** |
| Sensors | 5.76 | 3 | 4.61 | 11 |
| Breadboard | 3.5 | 1 | 3.5 | 1 |
| Arduino | 35.5 | 1 | 35.5 | 1 |
| Multiplexer | 1.2 | 1 | 1.2 | 1 |
| Clock Sensor | 1.3 | 1 | 1.3 | 1 |
| Total Cost (EUR) | 58.8 | | 92.1 | |

Information presented in Table 7 shows that the total costs of the proposed monitoring systems are EUR 58.8 for SHT35 and EUR 92.1 for SHT21. These costs are similar to commercial alternatives presented in the Table 5 with 0.5 °C accuracy (such as EL-USB-2 LASCAR (EUR 50 to 100)) and lower than other thermometers with the same accuracy (such as FLUKE 971 (EUR 350 to 500)). The assembling and programming cost of the monitoring system is not included in Table 7. Depending on the sensor type, the time is 10 min on average per sensor. Coefficient of variation ($R^2$) of the linear regression line between sensor measurements and reference data has been frequently used in the literature to verify the linearity of responses of low-cost sensors [60,61]. In other words, this parameter illustrates how well the measurements of studying sensors agree with the reference values, according to a regression module. A higher R-squared presents an increase in precision of the regression model. This analysis was carried out for the most inaccurate sensor arrangements and is summarized in Figure 14, where the pairwise correlation between sensor measurements and the associated statistical reference for an increasing number of sensors (ranging from one to three) are presented. According to Table 4, the most inaccurate sensor arrangements associated with combinations of one, two, and three units of sensors were selected among from 30, 435, and 4060 possible configurations, respectively. Evaluating the accuracy, the mean value of and individual sensor arrangement were compared with the associated statistical reference. For the combination of each number of sensors, the dependent (sensor data) and independent (statistical reference data) variables were plotted in Figure 14 on the vertical and horizontal axis, respectively. Comparison of $R^2$ values in an individual sensor set demonstrates how effective it is to increase the number of sensors. According to Figure 14, DHT22 provides the lowest linearity for the combination of all the studied number of sensors. For instance, for the case of one sensor, $R^2$ is equal to 0.9882 and reaches 0.9926 when the number of sensors is increased to three. However, when considering a number of sensors up to three, sensors SHT21 and SHT35 provided the best results, as $R^2 = 0.9992$ was obtained. Further assessment of the sensors in this figure also proves that the correlation of BMP180 pressure sensors with the associated statistical reference is strong, as the range of $0.9906 \leq R^2_{BMP180} \leq 0.9961$ was obtained when the number of sensors was increased from one to three.

The algorithm in Section 2.2.2 was applied to determine the accuracy of the best sensor combinations for each monitoring system, and the obtained results are summarized in Figure 15. In this figure, the horizontal axis represents the number of sensors considered in different arrangements (ranging from 1 to 8), and the vertical axis shows the absolute minimum error (Abs (Min Error)) generated by all possible combinations, as defined in Table 3.

The analysis of Figure 15 illustrates how increasing the number of sensors reduces the minimum errors significantly. This beneficial effect can be observed in the BMP280 monitoring system, as its minimum absolute error is reduced from 0.17 °C for one sensor to 0.006 °C for eight. Figure 15 additionally shows that the most accurate single low-cost sensors SHT21 (0.06 °C) and SHT35 (0.05 °C) are more accurate than the most accurate commercial thermometer (EXTECH EN510 (0.1 °C)).

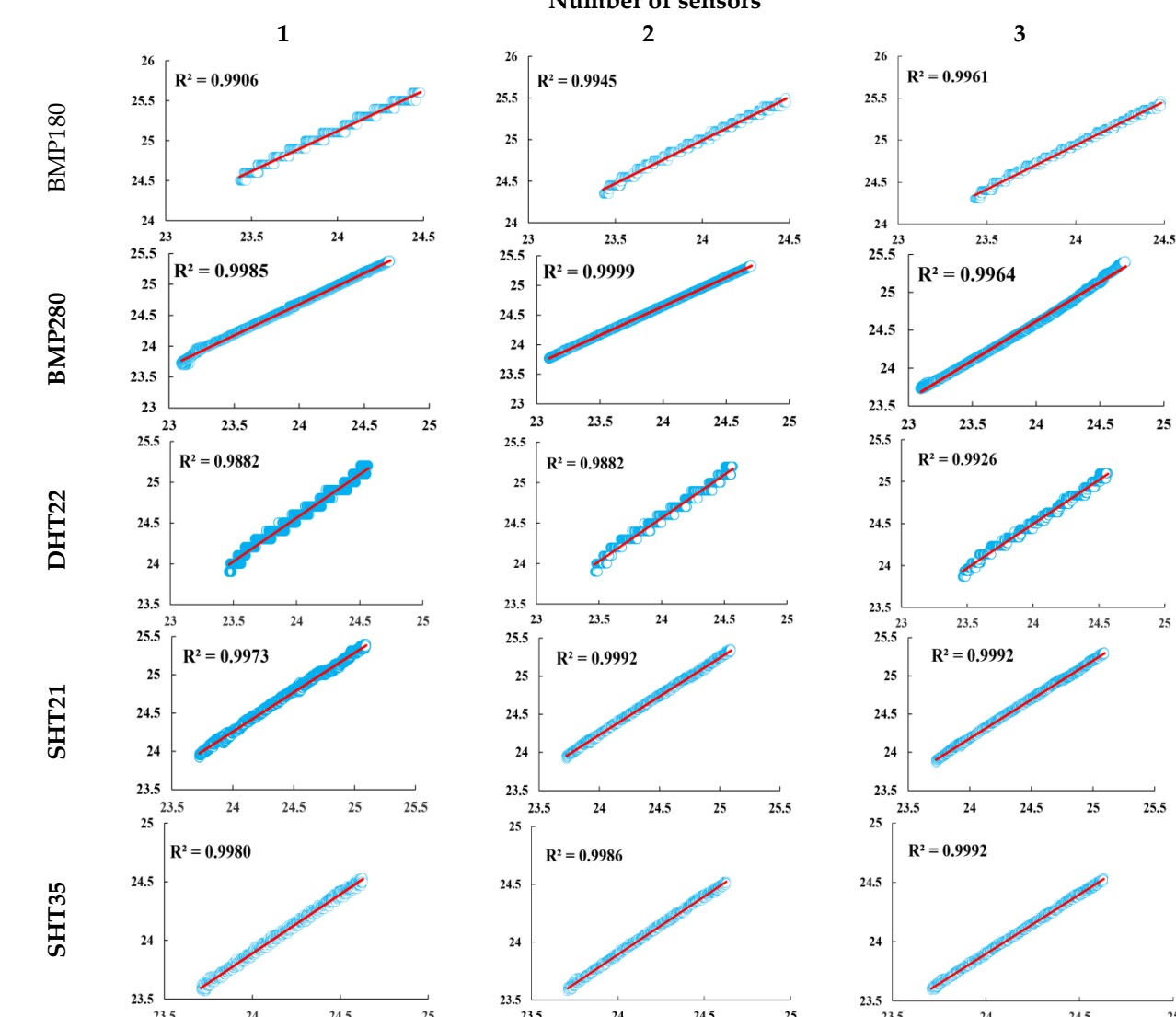

**Figure 14.** Pairwise correlations between all sensor sets and the associated statistical reference for an increasing number of sensors (ranging from 1 to 3). $R^2$ values were derived by the least-square regressions.

Nevertheless, to perform a cost comparison with the most accurate commercial thermometers, three sensors were considered in each monitoring system to enable multiple simultaneous temperature measurements. In this comparison, the two sensors that presented higher accuracy were BMP280 and SHT35. Their accuracies (0.015 and 0.023 °C, respectively) are significantly better than that of the best commercial thermometer reviewed (0.1 °C).

The cost of the different elements (sensors, microcontroller, breadboard and multiplexer) of aforementioned alternatives (SHT35 and BMP280) is detailed in Table 8.

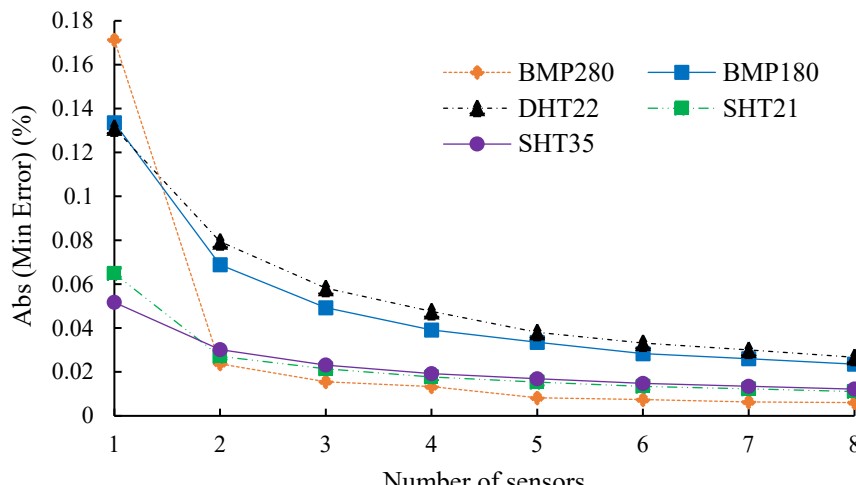

**Figure 15.** Minimum absolute errors between all possible arrangements of different sensors in the five developed monitoring systems.

**Table 8.** Cost of the chosen monitoring systems with the most accurate sensors of SHT35 and BMP280.

| | SHT35 | | BMP280 | |
|---|---|---|---|---|
| **Components** | **Price (EUR)** | **N°** | **Price (EUR)** | **N°** |
| Sensors | 5.7 | 3 | 3.6 | 4 |
| Breadboard | 3.5 | 1 | 3.5 | 1 |
| Arduino | 35.5 | 1 | 35.5 | 1 |
| Multiplexer | 1.2 | 1 | 1.2 | 1 |
| Clock Sensor | 1.3 | 1 | 1.3 | 1 |
| Total Cost (EUR) | 58.6 | | 55.9 | |

The information presented in Table 8 indicates that the total costs of the proposed monitoring systems are EUR 58.6 and 55.9 for SHT35 and BMP280, respectively. These costs are lower than those of the commercial alternatives (such as EXTECH EN510 (EUR 180 to 200) and TESTO 435-3 (EUR 750 to 1200) presented in Table 5; they also presented higher accuracies. As in the case of Table 7, the assembly and programming costs of the monitoring systems are not included in Table 8. The prices presented in Tables 7 and 8 are related to purchases carried out in September 2019 from an electronic store in Spain [62]. To better accomplish a comparison of the sensors presented in Table 4 with the proposed ones (Table 8) in this study, Table 9 is presented. This table contains the cost and technical characteristics of both series of sensors in terms of range and accuracy. It can be clearly seen that by adopting the proposed data analysis algorithm (in the Section 2.2.2.) to the measurements of the established monitoring systems, accuracies of 0.023 and 0.015 have been achieved for three sensors of each SHT35 and BMP280, respectively. When comparing prices of the proposed sensor sets with those of the commercial thermometers, a high level of cost efficiency could be observed. As installation of the sets SHT35 and BMP280 costs less than EUR 60, however, most of the commercial ones are higher than EUR 170. The development of accurate and affordable monitoring systems plays an important role in the quality of monitoring projects, especially for the ones with an extended period. Thus, simply by increasing the number of measurements (e.g., low-cost sensors) and at the same time adopting statistical analyses to the obtained data could highly help to carry out an efficient monitoring campaign.

**Table 9.** Comparison of the sensor characteristics.

| Name | SHT35 | BMP280 | EXTECH EN510 | TESTO 435-3 | FLUKE 971 | EL-USB-2 LASCAR |
|---|---|---|---|---|---|---|
| Accuracy (°C) | 0.023 | 0.015 | 0.1 | 0.2 | 0.5 | 0.5 |
| Range (°C) | [−40 to 125] | [−40 to 85] | [−100 to 1300] | [−25 to 75] | [−20 to 60] | [−35 to 80] |
| Price (EUR) | 58.8 | 55.9 | 180 to 220 | 750 to 1000 | 350 to 500 | 50 to 100 |

## 5. Conclusions

In this article, a methodology to improve monitoring projects has been proposed through a novel monitoring system. This monitoring system has proven the importance/advantage of increasing the number of sensors/measurements in monitoring projects. To validate performance of the proposed monitoring system, indoor temperatures have been studied experimentally through five developed monitoring systems, and the results have been compared with the associated statistical references and a traditional commercial thermometer. The comparison of three sensors sets, namely, 10, 20 and 30 sensors, has proven that the higher the number of sensors considered, the lower the standard deviation of the measurement, as evidenced by the standard deviation of 10 sensors (0.42) that decreased to 0.32 and 0.29 for the cases of 20 and 30 sensors, respectively. From the results, it can also be concluded that sensors SHT35 and BMP280 have the lowest ($0.028 \leq SD_{SHT35} \leq 0.066$) and highest ($0.336 \leq SD_{BMP280} \leq 0.404$) standard deviation ranges. The proposed algorithm has also indicated that the error information in the sensor catalogues does not correspond to the actual performance of these devices on site (the maximum obtained error of the BMP180 sensor (4.68%) was 57% higher than that presented in its catalogue). Moreover, the accuracy of standard commercial thermometers, such as the FLUKE 971 and EL-USB-2 LASCAR (0.5 °C), has been achieved by averaging eight sensors of SHT21 (concretely, 0.47 °C). The results additionally show that the proposed monitoring system can save up to 88% on costs compared to the most expensive commercial thermometers when choosing the best combinations of sensors. Future studies should address wireless communication protocols and on-site demonstration of sensor kits for long-term thermal and structural monitoring of buildings.

**Author Contributions:** Conceptualization, B.M. and J.A.L.-G.; methodology, F.J.C.P.; software, B.M.; validation, B.M. and S.K.; formal analysis, B.M. and S.K; investigation, B.M.; resources, B.M.; data curation, S.K. and B.M.; writing—original draft preparation, B.M.; visualization, S.K.; supervision, F.J.C.P.; project administration, J.A.L.-G.; funding acquisition, J.A.L.-G. All authors have read and agreed to the published version of the manuscript.

**Funding:** This research was funded by the Spanish Ministry of Economy and Competitiveness (grant number BIA2013-47290-R, BIA2017-86811-C2-1-R, and BIA2017-86811-C2-2-R) and by the Universidad de Castilla La Mancha (grant number 2018-COB-9092).

**Institutional Review Board Statement:** Not applicable.

**Informed Consent Statement:** Not applicable.

**Data Availability Statement:** No new data were created or analyzed in this study. Data sharing is not applicable to this article.

**Acknowledgments:** The authors are indebted to the Spanish Ministry of Economy and Competitiveness for the funding provided through the research projects BIA2017-86811-C2-1-R, directed by José Turmo and BIA2017-86811-C2-2-R, directed by José Antonio Lozano-Galant. All these projects are founded with FEDER funds. It is also to be noted that funding for this research has been provided to Behnam Mobaraki for his Ph.D. program by the Spanish Agencia Estatal de Investigación del Ministerio de Ciencia Innovación y Universidades (grant number: 2018-COB-9092).

**Conflicts of Interest:** The authors declare no conflict of interest.

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
