# Peer review of "Application of Low-Cost Sensors for Accurate Ambient Temperature Monitoring"

_buildings, doi:10.3390/buildings12091411_

Round 1
Reviewer 1 Report
This paper presents a Low-cost Sensors for Accurate Ambient Temperature Monitoring method for civil engineering use. The key of the technology is "low cost" and "equally quality" compared with existing sensors. The topic is very interesting and of benefit to the related induscty. The topic is very interesting and of benefit to the related induscty. The paper is conducted in a logic and clear way making it easy for readers to catch the purpose of the paper. The reviewer would like to simply provide the following comments to potentially improve the paper.
1. The low cost sensor clearly is applicable to buildings, bridges, tunnels and other infrastrctures. Why take "buildings" only in the title?
2. Better to create a summarized table in the end of the paper to overall compare the cost and technical paramters between proposed low cost sensors and existing regular sensors.
3. When referring to exact price number, currently in € in the paper, better add retail conditions like sources, year. The price varies a lot when the paper is released, which may mislead readers.
Author Response
Dear Reviewer,
Many thanks for all the comments on the submitted article. The authors have implemented all the comments to the updated version of the the article. Indeed the comments helped us to improve the quality of the article. Please find the attachment.
Sincerely yours
Behnam Mobaraki
Reviewer 2 Report
The paper presents a study dedicated to the development of a new low cost system. The introduction explicate the importance of the topic in the literature review. Anyway, it is not Clear the methodology for finding the scientific literature described in this section, and summarized in figure 1. the European research project MSCA Hello! Is dedicated to the development of low costs systems for indoor monitoring in Lacks, see https://doi.org/10.3390/en13112950 that introduce a low cost developed monitoring chamber for termostatare monitoring. In this research you can also find a paper on the Development of a Compatible, Low Cost and High Accurate Conservation Remote Sensing Technology for the Hygrothermal Assessment of Historic Walls based on Arduino. To have a comprehensive literature review some data lacks, such as keywords, number of papers found in different steps of the research. The methodology of the paper is not clear. Figure from 2 to 7 can be resumed in a Table, explicating the difference among systems. The development of the deviance is not clear, indicate better the reasons for selecting single systems. The explication of the experiment is clear, but it’s validation require a more accurate assessment.
Author Response

(The authors gave the same response as above.)

Round 2
Reviewer 2 Report
-